# Behind the Scene: Exploiting MC1R in Skin Cancer Risk and Prevention

**DOI:** 10.3390/genes12071093

**Published:** 2021-07-19

**Authors:** Michele Manganelli, Stefania Guida, Anna Ferretta, Giovanni Pellacani, Letizia Porcelli, Amalia Azzariti, Gabriella Guida

**Affiliations:** 1Department of Basic Medical Sciences, Neurosciences and Sense Organs, University of Bari-“Aldo Moro”, 70125 Bari, Italy; m.manganelli@unibs.it (M.M.); anna.ferretta@uniba.it (A.F.); 2DMMT-Department of Molecular and Translational Medicine, University of Brescia, 25123 Brescia, Italy; 3Department of Surgical-Medical-Dental and Morphological Science with Interest Transplant-Oncological and Regenerative Medicine, University of Modena and Reggio Emilia, 41124 Modena, Italy; stefania.guida@unimore.it; 4Department of Clinical Internal, Anesthesiological and Cardiovascular Sciences, Dermatology Clinic, Sapienza University of Rome, 00161 Rome, Italy; giovanni.pellacani@uniroma1.it; 5Laboratory of Experimental Pharmacology, IRCCS Istituto Tumori Giovanni Paolo II, 70124 Bari, Italy; l.porcelli@oncologico.bari.it (L.P.); a.azzariti@oncologico.bari.it (A.A.)

**Keywords:** melanocortin 1 receptor, MC1R, melanoma, basal cell carcinoma, squamous cell, skin cancer prevention

## Abstract

Melanoma and non-melanoma skin cancers (NMSCs) are the most frequent cancers of the skin in white populations. An increased risk in the development of skin cancers has been associated with the combination of several environmental factors (i.e., ultraviolet exposure) and genetic background, including melanocortin-1 receptor (*MC1R*) status. In the last few years, advances in the diagnosis of skin cancers provided a great impact on clinical practice. Despite these advances, NMSCs are still the most common malignancy in humans and melanoma still shows a rising incidence and a poor prognosis when diagnosed at an advanced stage. Efforts are required to underlie the genetic and clinical heterogeneity of melanoma and NMSCs, leading to an optimization of the management of affected patients. The clinical implications of the impact of germline *MC1R* variants in melanoma and NMSCs’ risk, together with the additional risk conferred by somatic mutations in other peculiar genes, as well as the role of *MC1R* screening in skin cancers’ prevention will be addressed in the current review.

## 1. Introduction

Skin cancers represent the most frequent cancer, with 5 million of new cases each year [1]. The most common skin cancers include melanoma and non-melanoma skin cancers (NMSCs), among which basal cell carcinoma (BCC) and squamous cell carcinoma (SCC). For the last 10 years the incidence of melanoma has been continuously rising, together with that of NMSCs, although a precise estimation of the number of NMSCs is impaired by the fact that their reporting in cancer registries is not mandatory [1,2]. While BCC is usually confined to the skin, SCC and melanoma can be potentially metastatic, being difficult to treat and therefore presenting with a poor prognosis [3,4].

An increased risk of skin cancers has been associated with a combination of environmental agents, such as ultraviolet radiation (UVR), and genetic background. In particular, genes involved in pigmentary regulation, such as melanocortin-1 receptor (MC1R), are implicated in skin cancers’ development [5,6].

The *MC1R* gene codifies for a G protein-coupled receptor (GPCRs), with a high affinity for α-melanocyte stimulating hormone (α-MSH). It is a highly polymorphic gene and it has been related to pigmentary as well as to non-pigmentary functions, including DNA repair.

In the last few years, there has been increasing knowledge about the MC1R functions and their clinical impact in dermato-oncology. The aim of this review is to provide an update on the impact of *MC1R* gene in melanoma and NMSCs’ risk—together with the additional risk conferred by somatic mutation of other gene—as well as its role in the prevention of skin cancers.

## 2. MC1R Structure, Regulation and Functions

### 2.1. MC1R Structure and Regulation

The *MC1R* gene (16q24.3, OMIM #155555) codes for seven transmembrane GPCRs of 317 amino acids, evolutionarily conserved [7,8], showing an extracellular N-terminus, with a glycosylation site, seven transmembrane segments, and an intracellular C-terminal extension including a palmitoylation site [9,10,11]. This receptor was first isolated from melanocytes, where its main physiological role in the ski has been shown [12]. MC1R shows a high affinity for the α-MSH, as well as for adrenocorticotropic hormone (ACTH) [9,10].

The *MC1R* gene may exhibit splice variants, giving rise to two forms of intergenic splicing, yielding *MC1R-TUBB3* (β-tubulin III) chimera and at least two forms of alternative splicing [13,14]. In all cases, the proteins encoded by the non-canonical mRNAs preserve the general architecture of GPCRs and differ from canonical MC1R for a longer C-terminal extension [14,15]. Once *MC1R* mRNA is translated, the receptor undergoes post-translational modifications that include oligomerization, N-glycosylation (Asn15, Asn29), palmitoylation (Cys315) and phosphorylation (Thr157, Thr308, Ser316) [11,16,17,18,19], contributing to receptor structure, localization, trafficking, internalization, desensitization.

Human MC1R shows a constitutive activation of downstream signaling, independent from the presence of the agonist, which is impaired in presence of MC1R variants [20]. Additionally, MC1R signaling is induced upon stimulation of human melanocytes with α-MSH [21], 12-O-tetradecanoylphorbol ester (TPA) [22] and [Nle^4^, DPhe^7^]-α-MSH (NDP-MSH, synthetic analog of α-MSH) [23]. The induction is also mimicked by the adenylyl-cyclase activator forskolin. Conversely, Agouti Signaling Protein (ASIP) inhibits α-MSH binding to MC1R [24,25]. Human β-defensin 3 (HBD3) prevents the binding of both α-MSH and ASIP to MC1R [26,27], preventing both the increase in cAMP and the upregulation of TYR in melanocytes.

Interestingly, paracrine factors produced by keratinocytes such as endothelin 1 (EDN1) and basic fibroblast growth factor (bFGF), act through their corresponding receptors on the plasma membrane of melanocytes to increase proliferation and differentiation. EDN1 mediates a dose-dependent upregulation of *MC1R* mRNA in normal human melanocytes [28], while the effects of bFGF are less clear, although an upregulation has been reported [28]. Interleukin-1-α (IL-1α) and interleukin-1-β (IL-1β) upregulate *MC1R* mRNA in normal human melanocytes [23], while TNF-α [29] and TGF-β [30], that potently repress melanogenesis in melanoma cells, moderately downregulate *MC1R* expression in normal melanocytes [23] and mouse melanoma cells [31].

Additionally, a downregulation of cyclic-adenosine monophosphate (cAMP) signaling has been related to: an increase in phosphodiesterase 4D3 gene (PDE4D), a transcriptional target of cAMP via microphthalmia-associated transcription factor (MITF) [32], limiting cAMP accumulation; the phosphatase and tensin homolog deleted on chromosome 10 (PTEN) [33,34] and the RING Finger domain-containing E3 ubiquitin ligase Mahogunin Ring Finger 1 (MGRN1) [35], inhibiting cAMP signaling, most likely due to a physical interaction of MGRN1 and MC1R; activation of ERK signaling, leading to MITF phosphorylation and degradation [36]; prolonged exposure to agonists, such as β-arrestins (ARRB), leading to desensitization or internalization of MC1R [17,27,37,38].

### 2.2. MC1R and Pigmentation

MC1R has a pivotal role in pigmentation, although important non-pigmentary functions have also been identified [12].

The pathway leading to pigmentation has been widely characterized. It involves α-MSH binding to the MC1R, with a consequent increase in adenylyl-cyclase (AC) activity and rising of intracellular cAMP levels. cAMP activates protein kinase A (PKA), leading to the phosphorylation of the cAMP-response element binding protein CREB, which in turn activates the promoter of the *MITF*. The resultant event is the upregulation of tyrosinase (TYR) and Tyr-related proteins (TYRP1 and DCT) (Figure 1), switching pheomelanin to eumelanin synthesis [36,39,40,41,42,43,44,45,46,47] (Figure 1).

The tyrosine-protein kinase c-KIT also plays a role in pigmentation [48,49,50,51]. Binding of the stem cell factor (SCF) to c-KIT induces sequential events [52,53], leading to recruitment of adaptor proteins containing a Src homology 2 (SH2) domain, which will associate with a guanine nucleotide exchange factor (GEF). The SH2/GEF complex activates RAS/RAF/MAPK/ERK cascade which, in turn, activates MITF [54,55,56,57] (Figure 1).

Indeed, the functional impairment of MC1R downstream signaling is characterized by prevalent red/yellow pheomelanin. Pheomelanin has weak shielding capacity against UVR relative to eumelanin and has been shown to amplify UVA-induced reactive oxygen species (ROS). Thus, an increased *ratio* of photoprotective eumelanins to pro-oxidant pheomelanins provides an effective shield against mutagenic UVR [58]. Moreover, considering the link existing between α-MSH and PPAR-γ (Peroxisome Proliferator-Activated Receptor γ), it has been shown that specific PPAR-γ modulators provide photoprotective effect in keratinocytes harboring MC1R-inactivating variants [59,60,61].

### 2.3. MC1R, Non-Pigmentary Functions and DNA Repair

Non-pigmentary functions of MC1R mediated via the α-MSH/MC1R pathway include modulation of pro-inflammatory cytokines [62], increasing matrix metalloproteases (MMPs), expression of adhesion molecules [63,64,65,66,67], increasing cellular energy production, liver and brown adipose tissue metabolism [67,68] and detoxification of ROS [69,70,71,72]. The cAMP pathway, through MITF, also activates the expression of the peroxisome proliferator-activated receptor γ coactivator-1α (PGC-1α), the master regulator of mitochondrial biogenesis [73,74,75]. Furthermore, MC1R contributes to melanocyte survival through the maintenance of genomic stability [76,77,78].

UVR includes UVA (320–400 nm) and UVB (290–320 nm), with UVA penetrating deep into the skin, reaching the dermis, inducing pigmentation by oxidation or distribution of pre-existing melanin, and UVB acting at epidermal level, inducing skin pigmentation through increased melanin synthesis [79]. UVR is one of the main DNA-damaging environmental factors [80]. The most predominant damage caused by UVB, eliciting alterations at epidermal level, is covalent linkage between two adjacent pyrimidines, resulting in cyclobutane pyrimidine dimer (CPD) and 6-4 photoproduct (6-4PP). The so-called “UVB signature mutations” are characterized by CT→T and CC→TT transitions [81,82,83,84,85]. UVA, penetrating deeper into the skin, reaching the dermis, induce oxidative stress, producing 7,8-dihydro-8-oxyguanine (8-oxodGuo) DNA damage, resulting in G-C to T-A transversion mutations [86,87,88,89]. UVA also efficiently promote photoisomerization of 6-4PPs into Dewar valence isomers [90,91]. Interestingly, α-MSH levels increase following UVR exposure [23,92,93,94,95,96,97], therefore contributing to the activation of the downstream signaling, leading to the modulation of the nucleotide excision repair (NER) pathway to enhance genomic stability and melanocytes resistance to UVR-mediated apoptosis [10,98,99,100,101,102,103,104,105]. MC1R activation by α-MSH triggers cAMP cascade and mediates on one hand the increase in PCNA (proliferating cell nuclear antigen) protein levels and on the other hand the phosphorylation of ATM and ATR (serine/threonine kinases). ATM and ATR, in turns, activate CHK1/2 (checkpoint serine/threonine kinases) [106] and promote the formation of γH2AX (histone H2Ax) [107,108], leading to the clearance of CPD and 6-4PP photoproducts. Moreover, ATM and ATR mediate the increase in DDB2 (Damage Specific DNA Binding Protein 2) and XPC (xeroderma pigmentosum, complementation group C). Additionally, PKA-dependent ATR phosphorylation, occurring independently from MITF, recruits XPA (xeroderma pigmentosum, complementation group A) to the sites of the photodamage, promoting DNA repair [107,108,109,110,111], together with NR4A2 (Nuclear Receptor Subfamily 4 Group A Member 2), XPC and XPE (xeroderma pigmentosum, complementation group E) [108,112,113] (Figure 2). α-MSH also enhances the expression of Base excision repair (BER) enzymes OGG1 (8-Oxoguanine DNA Glycosylase) and APE-1 (apurinic/apyrimidinic endonuclease 1) [72], and the phosphorylation of upstream activators of p53, ATR, and DNA-PK (DNA-dependent protein kinase) [72,103]. Activated p53 translocates to the nucleus to induce the expression of *p21* and *GADD45*, contributing to the repair of oxidative DNA damage [103,108,114,115].

## 3. *MC1R* Polymorphisms

*MC1R* is a highly polymorphic gene with more than 200 variants described to date [9,115,116,117]. Variant alleles including D84E, R142H, R151C, R160W and D294H have been defined as “R” or “RHC” alleles due to their strong association with the red hair color (RHC) phenotype [118]. The V60L, V92M and R163Q variants have a lower association with RHC phenotype and have been therefore designated as “r” alleles [9].

*MC1R* variants have been associated with a reduced receptor function, impairing the switch of melanin synthesis from eumelanin to the red–yellow pro-oxidant pheomelanin [115,118,119]. *MC1R* polymorphisms, both RHC and “r”, generate hypomorphic proteins, leading to different degrees of cAMP pathway activation impairment, therefore leading to reduced pigmentary and non-pigmentary MC1R functions, as described in the previous sections [120,121]. Specifically, D84E, R151C and R160W polymorphisms have been related to a decreased cell surface expression [122,123,124], due to deficient anterograde trafficking or increased desensitization and internalization [18,125,126], while an impaired coupling has been reported for R142H and D294H alleles [122,123,127]. Only a marginal effect of the V92M substitution on cell-surface expression or ability to activate downstream signaling has been reported [122,123].

## 4. Clinical Impact of *MC1R* Polymorphisms: Hair and Skin Color and Non-Invasive Imaging Features

Considering the pivotal role of *MC1R* in pigmentation, *MC1R* status has been proven to have a key role on hair and skin color. Homozygotes, compound heterozygotes as well as heterozygotes for RHC *MC1R* alleles have been associated with red hair color [128,129]. The V60L variant may act as a partially penetrant recessive allele. However, some individuals carrying compound heterozygote/homozygote *MC1R* variants do not have red hair. A possible explanation might be that red hair color has also been related to mutations in other genes (i.e., *POMC*) [130]. A dosage effect of *MC1R* variants on hair, as well as skin color, should also be considered, being implicated in different shades of red hair in heterozygotes as compared to homozygotes/compound heterozygotes. There is also evidence for a heterozygote effect on beard hair color, skin type and freckling [128], although an association between *MC1R* RHC polymorphisms and freckles has been demonstrated to be independent of skin and hair color [131]. A dosage effect of *MC1R* variant alleles on sensitivity to UVR has also been described [28]. Accordingly, heterozygotes for one variant allele show an intermediate ability to tan after repeated sun exposure between those with two variant alleles (most likely to be red hair subjects) and those with none of the variants. Therefore, a high frequency of *MC1R* heterozygous allele carriers could influence the skin’s response to UVR in most of the population who do not have red hair [132]. As a consequence, those who are homozygous/compound heterozygous for *MC1R* do not only have red hair, but also have pale skin, tan poorly and tend to burn on exposure to UVR, while subjects with pale skin who do not have red hair are more likely to be *MC1R* heterozygotes [128].

Non-invasive skin imaging performed with reflectance confocal microscopy (RCM) and optical coherence tomography (OCT), enabling in vivo evaluation of different layers of the skin, revealed a different dermal microenvironment in photoexposed skin of *MC1R* RHC variants carriers as compared to wild-type (WT) [133], suggesting a correlation between photoaging (aging related to UV exposure) in *MC1R* variant subjects and increased susceptibility to skin cancers [134]. Additionally, as revealed by clinical and daily routine non-invasive dermoscopy of nevi, *MC1R* status has an impact on nevus phenotype and RCM features. In detail, *MC1R* RHC variants carriers have a peculiar nevus phenotype, dermoscopically characterized by reduced structures and lower prevalence of atypical pigment network, visible vessels, dots and globules, and eccentric hyperpigmentation, associated with a high degree of skin freckling [135,136]. In addition, melanoma patients carrying *MC1R* variants as well as *CDKN2A* mutations, show clinically hypopigmented nevi and, at RCM, roundish cells infiltrating the dermo–epidermal junction [137].

## 5. *MC1R* and Skin Cancers Risk

Genome-wide association studies (GWAS) and meta-analyses have widely demonstrated the association of RHC variants with increased risk of melanoma [124,138,139,140,141] and NMSCs [71,124,131,142,143]. These associations were initially related to the pigmentary functions of *MC1R*, although many studies confirmed that the increased skin cancer susceptibility in *MC1R* carriers is independent from pigmentary traits [28,71,144,145,146].

### 5.1. MC1R and Melanoma Risk

Darker-pigmented subjects present a significantly higher risk of melanoma associated with *MC1R* variants [71,147,148,149,150,151,152]. Individuals carrying just one *MC1R* variant have almost 40% increased risk of melanoma, whereas carriers of two or more *MC1R* variants have more than a double risk, as compared to WT subjects [124,153]. In particular, the association of *MC1R* RHC variant alleles D84E, R142H, R151C, R160W and D294H with a direct effect on melanoma risk has been confirmed by several studies and meta-analyses [71,139,140,141,147,148,154,155]. However, melanoma in RHC individuals shows a significantly higher somatic mutational burden, as compared to melanoma patients without any RHC variants. Intriguingly, C > T and non-C > T were the most common mutations observed across all *MC1R* genotypes [156,157]. This might be related to a decreased protection against UVR damage in RHC carriers, or indicate that other mutational processes occur in melanocytes of these patients. Moreover, the number of *MC1R* variants also correlated positively with increased risk of melanoma development among individuals not showing the RHC phenotype [6,141]. A pooled analysis including 3830 melanoma cases and 2619 controls showed that the presence of any *MC1R* variant had a direct effect on melanoma, conferring a 60% higher risk to carriers versus non-carriers. Strikingly, considering the pigmentation-mediated effect of *MC1R* on melanoma risk prediction alone, it is smaller with any *MC1R* variant and each of the RHC and r variants [158]. Therefore, *MC1R* variants may partly mediate their effect through biological pathways that are independent of pigmentation and UVR [138,139,140,141,151].

A lower incidence and better survival rates for melanoma have been described in female subjects, as compared to males [159]. Interestingly, females carrying an RHC variant tended to exhibit significant lighter phototypes than males with the same *MC1R* genotypes, therefore contributing to different tanning ability between the two sexes.

Furthermore, *MC1R* variants have also been related to melanoma occurring in childhood and adolescents [160,161,162]. Interestingly, *MC1R* r variants were found to be more prevalent in childhood and adolescent melanoma than in adult ones, especially in patients aged 18 years or younger [163].

### 5.2. MC1R and NMSCs Risk

The most common NMSCs in fair-skinned populations are BCC and SCC [164]. UVR is the major environmental risk factor for NMSCs development [165], whereas fair skin and red hair are considered to be the most important phenotype risk factors [166]. Carriers of two *MC1R* variant alleles, mainly RHC variants, have a 2- to 3-fold increased risk of developing NMSCs, as compared to WT [131,153], also independently from phenotype [158].

In a study of 220 individuals (111 at high risk and 109 at low risk of BCC and SCC) in Queensland (area of high UVR), the prevalence of NMSCs was associated with the presence of *MC1R* RHC variant alleles R151C, R160W and D294H, whereas V60L, V92M and R163Q had minimal impact on BCC and SCC risk [167]. These findings were confirmed and extended in a case–control study of Dutch patients, showing the highest relative risks of NMSCs for D84E, H260P carriers, and slightly lower risks for R142H [131]. Data from the M-SKIP Project highlighted the association of *MC1R* variants and NMSCs development risk in populations living in different geographical areas, with a stronger role for darker-pigmented populations [70,124,131,143,168]. Interestingly, subjects with darker skin (skin types III and IV), carrying two *MC1R* variants, showed a lower risk of superficial multifocal BCC compared with *MC1R* variant carriers with lighter skin (skin types I and II), although the number of individuals in the analyzed subgroup was small [131].

The contribution of *MC1R* variants in the pathogenesis of each specific tumor type is not clear yet. Therefore, further investigation in functional studies focused on the carcinogenic mechanisms leading to BCC and SCC is needed [168].

## 6. *MC1R* Association with Melanoma Susceptible Genes

Germline *MC1R* variants may influence the mutational landscape of melanoma [146,156]. Despite the well-established impact of *MC1R* on skin cancer risk and development, the association of *MC1R* variants in combination with mutations in susceptible melanoma genes has not been clarified yet.

### 6.1. CDKN2A

An estimated 5–10% of all melanomas are hereditary, and of those, up to 40% are explained by germline mutations in cyclin-dependent kinase inhibitor 2a (*CDKN2A*). *CDKN2A* is the major susceptible gene in multiple primary melanoma patients [146,169]; it acts as a tumor suppressor gene, negatively regulating G1-S cell-cycle progression and promoting cellular senescence. Recently, a role for CDKN2A as a negative regulator of cellular oxidative stress has been suggested [170]. The first germline mutations in *CDKN2A* were reported in familial melanoma (V118D, G93W, R79P, N63S, R50Ter, IVS2 + 1 [G–T]) in 1994 [171]. Heterozygous loss of *CDKN2A* is sufficient to confer a 67% lifetime risk of melanoma [172] and it is associated with high inherited risk in melanoma prone families [173,174,175,176,177,178,179].

*MC1R* variants have been shown to increase the penetrance of *CDKN2A* mutations (observed risk over time for a mutation carrier), doubling the risk of melanoma development [124,154]. A stratified analysis of transmission of the R151C allele from parents to melanoma-positive offspring suggested that the contribution of the *MC1R* variant to the increased risk is independent of its effect on skin type [180]. Accordingly, a significant joint-effect of RHC variants (R163Q and D294H), considered either alone or in the presence of pigmentation and dysplastic nevi, influenced the penetrance of *CDKN2A* mutations in 20 French melanoma-prone families [181]. Helsing et al. reported that Norwegian melanoma patients showing both *CDKN2A* mutations and *MC1R* variants had an increased risk of melanoma when carrying D84E or R160W variants [182]. Additionally, carriers of A148T mutation of CDKN2A in association with non-synonymous MC1R variants (V60L, R151C, R160C and R163Q) have a 2- to 6-fold increased risk of melanoma [155,183]. Furthermore, germline carriers of the *CDKN2A p16-Leiden* deletion mutation in a large collection of Dutch families showed an increased risk of melanoma in carriers of *MC1R* variant alleles, with the R151C allele explaining most of this association [184].

In Queensland a *CDKN2A* mutation in association with *MC1R* variants have a raw penetrance of 84%, with a mean age at onset of 37.8 years when compared with family members who carry a *CDKN2A* mutation alone [179]. Accordingly, *CDKN2A* mutation carriers with *MC1R* variants had a significant lower median age at melanoma diagnosis than *CDKN2A* mutation carriers with no *MC1R* variants (37 years versus 47 years) [154,173,185]. Indeed, CDKN2A G101W mutation and *MC1R* variants carriers were younger at the first diagnosis with respect to WT multiple melanoma patients, showing hypopigmented nevi and roundish cells infiltrating the junction, suggesting an influence of *CDKN2A* mutation and *MC1R* variants in the development of dysplastic melanocytic lesions [137].

### 6.2. BRAF

Approximately 50% of melanomas harbor *BRAF* mutations. The association of *MC1R* variants with *BRAF kinase proto-oncogene* somatic mutations has been investigated in melanoma, showing different results among several populations.

An association between germline *MC1R* variants and somatic *BRAF* mutations was reported in tumors from United States and Italian populations. Carriers of at least one *MC1R* variant have a 5- to 15-fold increased risk confined only to *BRAF+* melanomas, regardless the presence of chronic solar damage signs. On the contrary, no association with *BRAF-* melanomas was reported, suggesting that people carrying germline *MC1R* variants have a greater risk of developing a melanoma harboring a *BRAF* mutation without skin photodamage [186,187].

The association of *MC1R* variants (independently from the number of variants [188]) with somatic *BRAF* mutations has not been replicated in Italian [189,190], Spanish [191], German [192], Australian [193] and North Carolina [194] populations. These conflicting findings across different populations have been related to a different distribution of *MC1R* variants among study populations or a risk-modifying effect due to sun exposure.

Interestingly, a negative association between *MC1R* variants and *BRAF* mutations has been described for head/neck melanomas, suggesting a difference in the pathogenesis of melanomas located at different skin sites, head/neck or trunk, which could contribute to their divergent prognoses [195]. Additionally, a low frequency of somatic *BRAF* mutations in RHC and non-RHC *MC1R* carriers was restricted to nodular melanoma [192].

### 6.3. Other Genes

*MC1R* polymorphisms in association with other susceptible genes for melanoma have also been reported. Kosiniak-Kamysz et al. detected significant intermolecular epistasis effects among *MC1R* and *TYR*, *SLC45A2* (solute carrier family 45 member 2) and vitamin D receptor gene *(VDR)*, with *MC1R* RHC variants and *TYR* rs1393350 (G > A) showing the highest statistical significance [196]. Only three studies focused on somatic mutations in the *TERT* gene promoter, all of which reported moderate-to-strong positive associations with *MC1R* variants [189,190,197]. Several genetic interactions of *MC1R* in melanoma also include *ASIP* [198] and X-ray repair cross-complementing protein (*XRCC*) [70].

## 7. *MC1R* Association with NMSCs Susceptible Genes

The association of *MC1R* variants in combination with mutations in susceptible NMSCs genes has not been investigated so far. The only evidence comes from Liboutet et al. who reported the P1315L mutation frequency in *PTCH* (Protein Patched Homolog 1, component of the hedgehog signaling pathway) not to be significantly different between BCC patients carrying a *MC1R* variant and those that do not carry one, suggesting an independent effect of both genes on BCC risk [199].

Interestingly, interactions between *MC1R* and melanoma susceptibility genes have been investigated, with inconsistent results [199].

However, future studies might be able to find potential correlations between different pathways leading to different great variability in NMSCs’ aggressiveness, morphology and response to treatment.

## 8. Epigenetic Regulation of *MC1R*

Epigenetic factors such as DNA-methylation chromatin-remodeling events, as well as gene regulation through non-coding RNAs play an important role in the pathogenesis of skin cancers [200,201,202,203,204,205,206,207,208]. Interestingly, epigenetic regulation of *MC1R* expression in melanoma has been recently investigated [209]. A methylated CpG-island (CGI) has been identified on a *MC1R* region, proposed as a *MC1R* enhancer. This CGI has been shown to control *MC1R* expression, with a slight trend of increased methylation in melanomas showing homozygous RHC *MC1R*, as compared to WT and heterozygous tumors. Interestingly, unmethylated tumors had a significantly worse prognosis compared to methylated tumors, although the prognostic effect of *MC1R* CGI methylation has not been fully elucidated [209].

Epigenetic interactions have been recently identified in animal studies, showing some biological mechanisms underpinning their induction, such as dietary intake of cysteine, as well as miRNA targeting *MC1R* transcript, which may have an impact on receptor regulation [210,211].

Taken together, these data shed light on the complex regulation of *MC1R* and efforts should be made to fully elucidate epigenetic regulation of the receptor in humans.

## 9. *MC1R* and the Impact of Skin Cancer Genetic Testing

The well-established role of *MC1R* in skin cancer risk, which has been proven to be independent from skin phototype, highlights the importance of this gene in genetic testing and skin cancers’ prediction. However, whether the feedback about genetic risk status may contribute to an increased skin cancer awareness, therefore leading to sun avoidance/protection and self-skin examination, and to a reduction of skin cancer risk is currently under debate [212].

A randomized controlled trial enrolling 73 patients with high risk of developing skin cancers was conducted to compare the impact of a strategy of *CDKN2A/MC1R* counseling, giving test results (intervention group), to a strategy of not offering genetic counseling and test results (usual care) on behaviors and skin cancer risk awareness. Patients enrolled were white, mainly females and college-educated. In particular, just one half of the patients were in the interventional group. This study, limited by the number of subjects enrolled and by the fact that just three patients were positive to *CDKN2A/MC1R* mutations/variants, did not show a significant impact of genetic counseling and test results on sun protection behaviors [213]. Another study, a randomized clinical trial by Hay et al., enrolling 499 patients, focused on the interest in and uptake of *MC1R* testing in the general population, after offering information about advantages and disadvantages of the test. Most of the people enrolled were non-Hispanic white, and had a high school diploma or less. Just a few patients were at a high risk for skin cancers, although more than one half of patients experienced sunburns. Interestingly, college-educated non-Hispanic white were significantly more prone to read information about *MC1R* testing while subjects experiencing sunburns were significantly more likely to request the test [214]. Importantly, a recent study showed that people with lower health literacy skills or education may need support to understand genetic test results, while higher skills were related to reduced distress after receiving the results of *MC1R* testing [215].

This result underlines the importance of personalized education for the correct communication with patients with different skills, in order to avoid the distress that might be related to the knowledge of increased susceptibility to skin cancers based on genetic information. However, the impact of *MC1R* testing on skin cancer awareness and sun avoidance behavior has not been established yet.

Interestingly, previous studies explored the role of *MC1R* as a prognostic marker in metastatic melanoma and as a potential approach for target treatments in skin cancers. A significant correlation between *MC1R* variants and worse outcomes (overall response rate and progression-free survival) in *BRAF*-mutated metastatic melanoma patients treated with target therapy was observed, due to the interactions of *MC1R* with other pathways [216].

Additionally, considering that impaired function of MC1R has been related to an increased susceptibility to skin cancers, MC1R agonists and antagonists might be employed as a potential therapeutic approach [12,217]. Accordingly, in vitro and animal studies have shown that forskolin, through increasing cAMP levels, induced an improvement in NER function and DNA repair [108,218,219]. Furthermore, regulation of palmitoylation, which has been shown to be reduced in RHC MC1R, has been proven to reduce melanoma risk in in vitro and animal models [11,19].

However, MC1R has not been employed as a target treatment for skin cancers in humans so far. Currently, an α-MSH analogue is employed in clinical practice for treating photosensitivity in patients with erythropoietic protoporphyria [220].

Based on the information currently available in the literature, future studies are needed in order to provide data concerning the role of personalized genomic risk, including *MC1R*, with potential clinical impact in terms of early detection, treatment and preventive strategies.

## 10. Conclusions

As knowledge is expanding very rapidly, future directions should be addressed to determine the biological mechanisms underlying non-pigmentary *MC1R* functions, evaluate the gene–gene and gene-environment interactions, and to incorporate *MC1R* variants into melanoma and NMSCs risk prediction models and test their effect on motivating risk-reducing behaviors as a cancer prevention strategy. Additionally, non-invasive skin imaging evaluation, together with genetic studies, might improve the recognition of early skin variations preceding skin tumors’ development as well as melanoma and NMSCs’ identification and early tumor diagnosis, for a patient-tailored management protocol.

## Figures and Tables

**Figure 1 genes-12-01093-f001:**
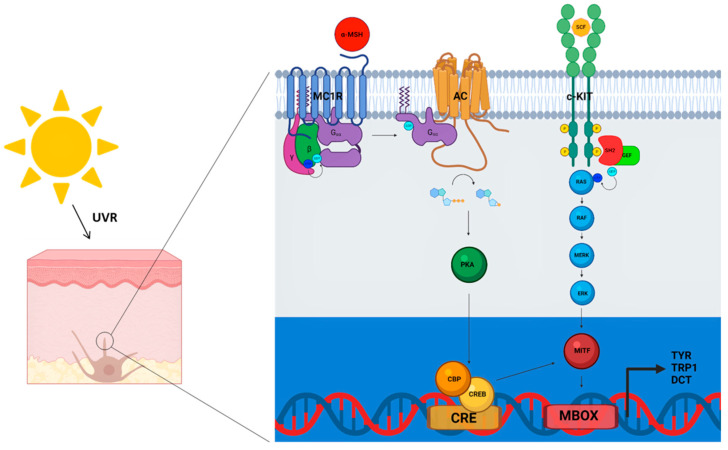
Pigmentary pathway of MC1R. Binding of α-MSH on MC1R receptor activates adenylyl cyclase (AC) and stimulates cAMP production, which in turns induce the activation of several downstream effectors, including MITF transcription factor. MITF binds the MBOX on the promoters of tyrosinase (*TYR*), phosphoribosylanthranilate isomerase (*TRP1*) and dopachrome tautomerase (*DCT*) genes, leading to the expression of different enzymes involved in melanin biosynthesis. Melanin acts as a UV-protective shield in the epidermis. (Gα-β-γ proteins, CREB (cAMP response element binding protein), CBP (CREB-binding protein), CRE (cAMP response elements), c-KIT (tyrosine-protein kinase KIT), SH2 (Src homology 2), GEF (Guanine nucleotide Exchange Factor), MAPK/ERK (extracellular signal-regulated kinases)). This figure was created with BioRender.com.

**Figure 2 genes-12-01093-f002:**
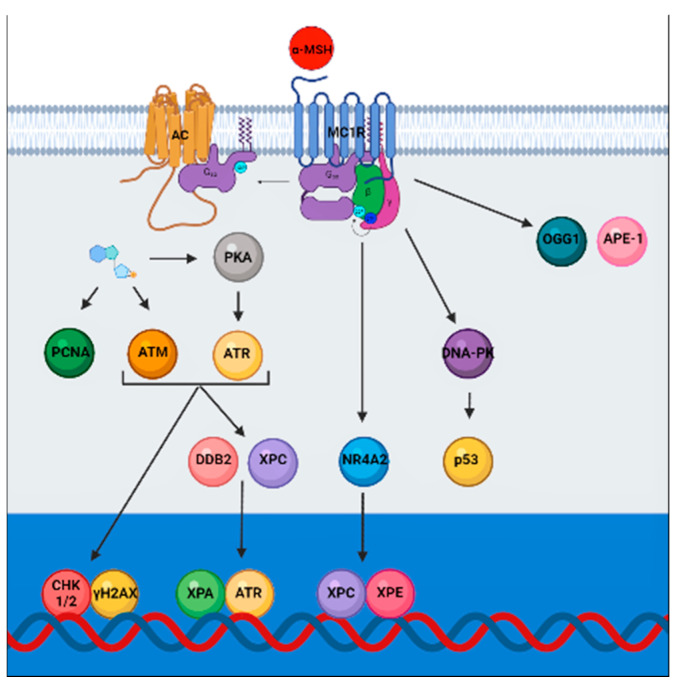
MC1R signaling promotes genomic stability through NER and BER. MC1R activation induces translocation of NR4A2 to the nucleus where it co-localizes with XPC and XPE at the sites of UV-induced DNA damage. MC1R activation also leads to elevated levels of XPC and γH2AX, promoting the formation of DNA repair-complexes. Levels of γH2AX are regulated downstream of ATR and by DNA-PK mediated phosphorylation of p53. In addition, PKA activation promotes the phosphorylation of ATR and ATR complexes with XPA in the nucleus. Following phosphorylation of XPA, the complex translocates to the sites of UV-induced DNA damage. α-MSH also enhances the expression of OGG1 and APE-1. The cumulative effects is maintenance of genomic stability of melanocytes by activation of nucleotide excision and base excision repair pathways (NER and BER). This figure was created with BioRender.com.

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
