# Peer review of "Behind the Scene: Exploiting MC1R in Skin Cancer Risk and Prevention"

_genes, 2021, doi:10.3390/genes12071093_

Round 1
Reviewer 1 Report
The manuscript entitled “Behind the scene: Exploiting MC1R in skin cancer risk and prevention” introduces the reader in a compact form, but dense in information, of the extensive MC1R functions. The authors describe the pigmentary pathways and non-pigmentary functions of MC1R. About the latter, they put the emphasis on DNA repair mechanisms. The figures are informative and concise giving a clear image of the mechanisms described. As clearly stated in the manuscript, MC1R plays a fundamental role against UV damage, and the authors describe the relationship of MC1R polymorphisms and skin cancer risk, hair and skin color under a cancer risk model. They also describe the MC1R association with other melanoma susceptibility genes like CDKN2A and BRAF (and others), giving a compact panorama of the topic. As a review of the topic, it gives a well documented update of MC1R and MC1R variants related to skin cancer. However, there is an item not mentioned explicitly and it is my curiosity. We will appreciate a section where epigenetic is taken into account. Recently, different papers have shown the relationship of DNA methylation and vitiligo, or how RHC variants are associated with the DNA methylation status of the surrounding genomic region. Also, animal studies found micro RNAs which targeted the transcript of MC1R, or the effect of cysteine in dietary and epigenetic changes in MC1R. Including an epigenetic section will help the readers to think in a more extended view of the topic.
Author Response
The manuscript entitled “Behind the scene: Exploiting MC1R in skin cancer risk and prevention” introduces the reader in a compact form, but dense in information, of the extensive MC1R functions. The authors describe the pigmentary pathways and non-pigmentary functions of MC1R. About the latter, they put the emphasis on DNA repair mechanisms. The figures are informative and concise giving a clear image of the mechanisms described. As clearly stated in the manuscript, MC1R plays a fundamental role against UV damage, and the authors describe the relationship of MC1R polymorphisms and skin cancer risk, hair and skin color under a cancer risk model. They also describe the MC1R association with other melanoma susceptibility genes like CDKN2A and BRAF (and others), giving a compact panorama of the topic. As a review of the topic, it gives a well documented update of MC1R and MC1R variants related to skin cancer.
However, there is an item not mentioned explicitly and it is my curiosity. We will appreciate a section where epigenetic is taken into account. Recently, different papers have shown the relationship of DNA methylation and vitiligo, or how RHC variants are associated with the DNA methylation status of the surrounding genomic region. Also, animal studies found micro RNAs which targeted the transcript of MC1R, or the effect of cysteine in dietary and epigenetic changes in MC1R. Including an epigenetic section will help the readers to think in a more extended view of the topic.
R: We are grateful to the referee for taking the time to revise our paper, highlighting its strengths.
In the revised version of the manuscript we added a small section on epigenetics, where we highlighted studies taking into account MC1R and DNA methylation and miRNA (10.1111/mec.15024; 10.3389/fgene.2020.00047), according to the suggestion of the reviewer and highly-citated papers in the journal (10.3390/genes10020102; 10.3390/genes10020172).
Reviewer 2 Report
The reviewed article presents the role of MC1R in the development of melanoma and nonmelanoma skin cancers. The Authors also described the structure, polymorphisms, signaling pathways, and physiological functions of MC1R. The topic of the paper is interesting and valuable in general. However, there are several issues to consider.
- line 19: “Melanoma and non-melanoma skin cancers (NMSCs) are the most frequently diagnosed cancers of the skin”: the sentence raises the question, what other skin cancers are.
- The Introduction section is surprisingly short and lacks more detailed information. I suggest presenting more detailed epidemiological data on melanoma and NMSCs. It is also worth introducing basic information on diagnostic problems and the effectiveness of treatment.
- The Authors did not describe what cellular or molecular factors influenced MC1R expression.
- line 54: “Increased TYR activity leads to the synthesis of black/brown eumelanin pigments, whereas functional impairment of MC1R downstream signaling is characterized by prevalent red/yellow pheomelanin.”: I cannot agree with the following statement. Tyrosinase is already involved in the initial stages of melanogenesis that are common to the synthesis of both eu- and pheomelanin. The proportion of eu - to pheomelanin depends on many biochemical factors. It is also related to the skin phototype, so it is an innate and physiological feature. Are the Authors sure that it is mainly about the activity of tyrosinase or maybe about the amount/content of the enzyme?
- Figures 1 and 2 show the activation of signal pathways involving MC1R. I suggest presenting AC activation by the a-subunit of G proteins.
- Figure 1 shows the signaling pathway with the c-KIT receptor and RAS kinases. This issue has not been described in the text of the paper.
- It is worth emphasizing the different biological effects of UVA and UVB radiation, also in the context of melanogenesis.
- line 122: “MC1R variants have been associated to a reduced receptor function, impairing the switch of melanin synthesis from eumelanin to the red-yellow pro-oxidant pheomelanin [48,56,57].”: The sentence is not entirely clear. What does a reduced receptor function mean? How is the switching of melanin synthesis disturbed?
- Could the Authors indicate the possibilities of using MC1R genetic variants in prognosis for patients with skin cancer and personalizing the therapy?
Author Response
The reviewed article presents the role of MC1R in the development of melanoma and nonmelanoma skin cancers. The Authors also described the structure, polymorphisms, signaling pathways, and physiological functions of MC1R. The topic of the paper is interesting and valuable in general.
R: we thank the referee for the comments and the help in improving the paper.
However, there are several issues to consider.
- line 19: “Melanoma and non-melanoma skin cancers (NMSCs) are the most frequently diagnosed cancers of the skin”: the sentence raises the question, what other skin cancers are.
R: The sentence has been modified in the text.
- The Introduction section is surprisingly short and lacks more detailed information. I suggest presenting more detailed epidemiological data on melanoma and NMSCs. It is also worth introducing basic information on diagnostic problems and the effectiveness of treatment.
R: The introduction section has been implemented.
- The Authors did not describe what cellular or molecular factors influenced MC1R expression.
R: Cellular and molecular factors influencing MC1R expression have been introduced in a dedicated section (mc1r structure and regulation)
- line 54: “Increased TYR activity leads to the synthesis of black/brown eumelanin pigments, whereas functional impairment of MC1R downstream signaling is characterized by prevalent red/yellow pheomelanin.”: I cannot agree with the following statement. Tyrosinase is already involved in the initial stages of melanogenesis that are common to the synthesis of both eu- and pheomelanin. The proportion of eu - to pheomelanin depends on many biochemical factors. It is also related to the skin phototype, so it is an innate and physiological feature. Are the Authors sure that it is mainly about the activity of tyrosinase or maybe about the amount/content of the enzyme?
R: We agree with the reviewer and the manuscript has been changed accordingly.
- Figures 1 and 2 show the activation of signal pathways involving MC1R. I suggest presenting AC activation by the a-subunit of G proteins.
R: Figures 1 and 2 have been modified according to the suggestion of the reviewer, presenting AC activation by the a-subunit of G proteins.
- Figure 1 shows the signaling pathway with the c-KIT receptor and RAS kinases. This issue has not been described in the text of the paper.
R: c-KIT receptor and RAS kinases cascade have been briefly described, according to the suggestion
- It is worth emphasizing the different biological effects of UVA and UVB radiation, also in the context of melanogenesis.
R: In the revised version of the manuscript UVA and UVB radiation, also in the context of melanogenesis.
- line 122: “MC1R variants have been associated to a reduced receptor function, impairing the switch of melanin synthesis from eumelanin to the red-yellow pro-oxidant pheomelanin [48,56,57].”: The sentence is not entirely clear. What does a reduced receptor function mean? How is the switching of melanin synthesis disturbed?
R: This issue has been addressed in the manuscript. Briefly, MC1R polymorphisms induce the synthesis of hypomorphic receptor, impairing the activation of downstream pathways as indicated in the section 2,2 and 2.3.
- Could the Authors indicate the possibilities of using MC1R genetic variants in prognosis for patients with skin cancer and personalizing the therapy?
R: In the revised version of the manuscript we included MC1R as a prognostic marker in metastatic melanoma and as a potential approach for target treatments in skin cancers, as suggested.
Round 2
Reviewer 2 Report
In response to the comments, the Authors improved the manuscript. They responded to the questions. The answers and introduced changes have dispelled the most problematic issues. The work has appeared to be more precise and accurate.